# The Role of Traditional and Religious Beliefs in HIV Testing and Prevention in Africa: A Scoping Review Protocol

**DOI:** 10.3390/ijerph22050743

**Published:** 2025-05-08

**Authors:** Silingene Joyce Ngcobo, Tracy Zhandire

**Affiliations:** Discipline of Nursing and Public Health, University of KwaZulu-Natal, Durban 4000, South Africa; ngcobos5@ukzn.ac.za

**Keywords:** HIV prevention, cultural beliefs, religious beliefs, scoping review protocol, Africa, HIV testing, HIV interventions

## Abstract

Human immunodeficiency virus remains a persistent public health challenge in Africa, with cultural and religious beliefs influencing testing and prevention behaviors. Understanding these influences is important for developing culturally appropriate interventions. This scoping review protocol outlines a plan to systematically map the existing literature on the impact of traditional and religious beliefs, and the roles of traditional and religious leaders, on HIV testing and prevention behaviors across African contexts. Following Arksey and O’Malley’s framework, the review will progress through six methodical stages. By identifying, analyzing, and summarizing the relevant literature and consulting stakeholders, this review aims to inform the development of culturally informed public health interventions and identify areas requiring further research. The intended outcome is a structured overview of how traditional and religious beliefs and leaders shape HIV testing and prevention behaviors, identifying contextual factors to guide the creation of effective HIV prevention programs within African communities. This scoping review protocol has been shared on Open Science Framework (OSF).

## 1. Introduction

Human immunodeficiency virus (HIV) continues to be a significant public health challenge in Africa, necessitating persistent efforts to enhance testing and preventive strategies [1]. In many African societies, cultural and religious frameworks significantly influence health-seeking behaviors [2,3]. Faith-based organizations (FBOs) and religious leaders play significant roles in disseminating HIV-related information, with churches and mosques serving as trusted spaces where health guidance is shared during sermons [4].

These religious institutions shape health perceptions and behaviors through multiple mechanisms: they establish moral frameworks that guide sexual behavior and partner selection; they provide pastoral counseling that influences health decisions; they implement faith-based health programs such as pre-marital HIV testing requirements; and they determine whether certain preventive measures (such as condom use) are endorsed or discouraged within their communities [5]. Additionally, the high level of trust placed in religious leaders means that their public statements about HIV can either reduce stigma or reinforce it, directly affecting community members’ willingness to seek testing and treatment [6].

While traditional and religious frameworks can support health-seeking behaviors, certain beliefs may also pose challenges to HIV testing and prevention. For instance, the virgin cleansing myth, the belief that sexual intercourse with a virgin can cure HIV/AIDS, has been reported in various African communities, leading to harmful practices and increased transmission risks [7]. Some individuals may prioritize spiritual or traditional remedies over biomedical interventions, potentially impacting adherence to HIV treatment regimens [8,9]. Local beliefs, rooted in historical, spiritual, and communal traditions, profoundly shape health perceptions and behaviors, varying across Africa’s diverse cultural landscapes [10].

Furthermore, cultural practices such as widow inheritance and sexual cleansing rituals, where widows are expected to engage in unprotected sex to purify themselves, have been identified as contributing factors to HIV spread [11]. While biomedical interventions have demonstrated effectiveness in clinical trials, their acceptance and implementation are often mediated by these cultural and religious contexts [12]. For example, in Gweru, Zimbabwe, some individuals living with HIV have discontinued antiretroviral therapy (ART) in favor of faith healing practices, believing that spiritual interventions can cure HIV, leading to treatment defaulting and posing challenges to HIV management efforts [13]. In Tanzania, certain religious beliefs contribute to HIV-related stigma; for instance, the perception of HIV as a punishment from God can lead to shame and reluctance to seek testing or disclose their HIV status, thereby hindering prevention and treatment efforts.

In addition to the previously mentioned challenges, specific religious beliefs can significantly impact HIV testing and treatment adherence. For example, in some Pentecostal Christian communities in sub-Saharan Africa, individuals believe that spiritual healing through prayer and the intervention of religious leaders can cure HIV. This belief may lead some to discontinue antiretroviral therapy (ART), viewing continued medication as a lack of faith in divine healing [14]. In Tanzania, reports indicate that certain faith leaders advise congregants to rely solely on prayer for healing, resulting in some individuals abandoning their treatment regimens [15]. Similarly, in Uganda, a study found that a subset of patients discontinued ART due to a belief in spiritual healing, highlighting the need for culturally sensitive adherence counseling that addresses such beliefs [16].

Despite their importance, the influence of traditional and religious beliefs on attitudes and behaviors related to HIV testing and prevention has not received equivalent attention for biomedical approaches [5]. This imbalance can be attributed to several factors. First, global health responses have historically prioritized biomedical interventions over socio-cultural factors due to their ease of standardization, quantifiable impact, and alignment with the dominant biomedical paradigm in international health programming [17,18]. Second, funders often favor interventions with measurable short-term outcomes, whereas addressing belief systems typically requires longer-term engagement with less immediately quantifiable results [19].

Third, the complexity and diversity of belief systems across different African communities present methodological challenges for researchers attempting to develop generalizable knowledge [20]. Finally, there has been a historical tendency in public health research to view traditional beliefs as barriers to overcome rather than as potential resources to incorporate into comprehensive prevention strategies [21]. As a result, research in public health has often under-represented qualitative studies that explore the nuanced role of belief systems in shaping health behaviors.

This scoping review aims to map the existing literature, identify knowledge gaps, and analyze the roles of religious and traditional leaders in shaping HIV-related behaviors. By providing insights into culturally sensitive strategies that align with local beliefs and practices, the findings will assist policymakers, public health professionals, and researchers in designing more effective HIV interventions across diverse African communities.

This review will examine the roles of traditional and religious leaders, identify cultural barriers and facilitators, and offer insights that could guide the development of contextually relevant HIV prevention strategies. By recognizing and understanding local beliefs, this study aims to inform more effective and community-driven interventions, thereby contributing to the sustainability and success of HIV prevention efforts across diverse African contexts.

The ultimate goal is to develop approaches that respectfully integrate biomedical interventions with local belief systems rather than positioning them in opposition. While this review provides a regional overview of Africa, it will also highlight variations between different communities and identify specific areas where further research may be beneficial, particularly studies focusing on distinct cultural contexts or comparative analyses across different African settings.

## 2. Materials and Methods

This scoping review will utilize the six-stage methodological framework developed by Arksey and O’Malley, as detailed in the Joanna Briggs Institute (JBI) Manual for Evidence Synthesis [22,23]. The framework consists of the following stages: (1) identifying the research question, (2) identifying relevant studies, (3) study selection, (4) charting the data, (5) collating, summarizing, and reporting results, and (6) the consultation phase. These stages ensure a systematic and comprehensive approach to mapping the existing literature on the role of traditional and religious beliefs in HIV testing and prevention. A scoping review conducted by Conteh, Latona and Mahomed [24] utilized Arksey and O’Malley’s six-stage methodological framework.

This review aims to systematically examine the existing literature to understand how traditional and religious beliefs influence individuals’ decisions regarding HIV testing and adherence to prevention strategies across Africa. It will also explore interventions that have successfully integrated cultural and religious considerations to enhance HIV prevention and testing uptake. By mapping current evidence, this study seeks to inform culturally sensitive approaches that respect and incorporate traditional and religious contexts in combating HIV/AIDS in African communities.

A systematic search will be conducted across multiple databases, including PubMed, Scopus, Web of Science, AJOL, Africa-Wide Information, CINAHL, PsycINFO, ERIC, LLBA, and Embase, using predefined search terms and inclusion criteria, focusing on peer-reviewed studies published in English from 2004 to the present. This timeframe will allow us to capture the most recent and relevant research, ensuring that the review reflects contemporary developments in HIV/AIDS prevention, testing, treatment, and the evolving cultural and religious contexts in African communities. Additionally, it covers the period of significant policy shifts and global health initiatives that have influenced HIV-related interventions, providing an up-to-date synthesis of evidence for culturally sensitive strategies in HIV prevention and testing.

The selection process involves two stages: an initial screening of titles and abstracts followed by a full-text review. Two independent reviewers will conduct these screenings, with disagreements resolved through discussion or consultation with a third reviewer. Data extraction will utilize a standardized charting form to capture the study design, population characteristics, and key findings. The results will be qualitatively summarized, grouping studies by relevant themes, with a narrative synthesis presenting the findings.

Consultations with healthcare professionals, researchers, and policymakers will refine the findings and ensure their practical relevance. This review is registered with the Open Science Framework (OSF), https://osf.io/8v5nf created on the 3rd of March 2025, and reporting will adhere to PRISMA-ScR guidelines. This study is scheduled from September 2025 to April 2026, encompassing all phases from data collection to reporting. While the primary aim is to map the literature’s breadth, the relevance of the studies will be rigorously evaluated during selection and synthesis, focusing on alignment with the research question, study design, and contextual factors.

Scope of the review

This scoping review aims to systematically examine and synthesize existing research on the following:Influence of traditional and religious beliefs: investigating how traditional and religious beliefs in Africa shape attitudes and behaviors toward HIV testing and prevention.Role of traditional and religious leaders: examining the involvement of traditional and religious leaders in either promoting or hindering HIV testing and prevention initiatives within their communities.

Focusing on Africa is essential due to its rich cultural diversity and the substantial impact of traditional and religious practices on health behaviors. Understanding these cultural contexts is crucial for developing interventions that resonate with local populations. By analyzing the existing literature on these topics, this review seeks to inform the development of culturally sensitive and effective HIV testing and prevention programs across diverse African communities.

### 2.1. Stage 1: Identification of the Review Question

The Population, Concept, and Context (PCC) framework was used to identify the main concepts in the primary review question and inform the search strategy (Table 1). This review addresses the following research questions and objectives.

This scoping review seeks to address the following primary research question:

Research questions

Individual-level beliefs—how do traditional and religious beliefs in Africa influence attitudes and behaviors toward HIV testing and prevention?Community-level leadership roles—what roles do traditional and religious leaders play in influencing HIV testing and prevention initiatives within their communities?

Objectives

To achieve this aim of this scoping review, the following objectives will be pursued:To establish the influence of traditional and religious beliefs on HIV testing and prevention in Africa.To investigate how traditional and religious beliefs in Africa influence attitudes and behaviors toward HIV testing and prevention.To examine how traditional and religious leaders influence HIV testing and prevention initiatives within their communities.To identify gaps in research on the intersection of cultural and religious beliefs and HIV testing and prevention and suggest future research directions.

Understanding these aspects is crucial for developing culturally appropriate and effective health education materials that resonate with local populations, thereby improving health literacy and outcomes (Table 1).

### 2.2. Stage 2: Identification of Relevant Studies

Search strategy

A comprehensive search strategy will be employed to identify studies on traditional and religious beliefs, the roles of traditional and religious leaders, and their influence on HIV testing and prevention behaviors in Africa. The search will cover multiple databases to ensure broad coverage, including PubMed, Scopus, Web of Science, AJOL, Africa-Wide Information, CINAHL, PsycINFO, ERIC, LLBA, and Embase. Tailored search strategies will be developed for each database using relevant keywords and Boolean operators.

To validate the search strategy, known seminal studies will be tested to ensure accuracy and comprehensiveness. Based on the results, search terms, inclusion/exclusion criteria, or database selections may be adjusted. The final search strategy will be shared openly on OSF, https://osf.io/8v5nf for transparency.

Additional search strategies

To enhance the comprehensiveness of the literature search, additional methods will be employed. Backward citation searching (ascendancy approach) will involve reviewing the reference lists of selected articles to identify seminal and foundational studies that may not have been captured in the initial search. Forward citation searching (descendancy approach) will be conducted using platforms such as Crossref, Google Scholar, and Scopus to track newer studies that have cited the included articles, ensuring that influential and recent research is incorporated. Additionally, co-citation analysis using CoCites will be performed to identify frequently co-cited articles, helping to uncover key studies that may not have appeared in the primary search but are influential within the field.

Search validation procedure

A pilot search will be conducted across selected databases to test the effectiveness of search terms and filters. Known relevant studies will be used for validation, and adjustments will be made as necessary to ensure comprehensive coverage.

Gray literature

Gray literature will be included to provide a well-rounded perspective. Sources will include:Preprints from servers such as ArXiv, PsyArXiv, and MedRxiv.Google Scholar searches for additional unpublished works.Dissertations and theses from ProQuest Dissertations & Theses Global.Conference abstracts and reports from organizations like WHO, UNAIDS, AVAC, and PEPFAR.

These sources will help to capture insights beyond traditional peer-reviewed literature.

Justification for search decision

The selected databases ensure broad coverage of health and social sciences, particularly African literature. Gray literature sources were included to incorporate unpublished and non-peer-reviewed findings that may offer valuable insights. Query strings will be designed to balance sensitivity and precision, capturing relevant studies while minimizing irrelevant results.

Screening, data management and sharing

EndNote 21 and Rayyan will be used for reference management, duplicate removal, and organizing the screening process. Screening will be conducted by two independent reviewers to ensure reliability and validity in study selection. Rayyan will also be used for data extraction and charting, providing systematic tracking and transparency in decision making.

For relevant non-English gray literature, professional translation services will be employed to ensure accurate data extraction. If study eligibility is unclear, authors will be contacted via email for clarification. Metadata related to author communications (e.g., contact details, dates, and responses) will be documented for transparency, with authors’ permission sought before sharing this information to protect their privacy.

Upon completion of the review, all data—including bibliographic details and screening decisions—will be shared via the Open Science Framework (OSF) repository. Files will be available in RIS, CSV, and XLSX formats for accessibility. The data will be openly available with no embargo, promoting transparency and reproducibility in the scoping review process.

Contacting authors for clarification

In this scoping review, authors will be contacted if there is ambiguity regarding their study’s eligibility or methodology, such as missing data or unclear findings related to traditional and religious beliefs on HIV testing and prevention in Africa. Initial contact will be made via email. If no response is received within two weeks, a polite follow-up email will be sent. Further contact will only be pursued if absolutely necessary.

Metadata of all communications, including author names, contact details, dates of contact, and responses, will be documented for transparency. Authors will be asked for permission to share these metadata, with assurance that responses will be used solely for the review and that no personal information will be disclosed without consent.

This process will be detailed in the methodology section to ensure transparency. Data on author responses and clarifications will be presented in a concise table or narrative format, allowing readers to assess the completeness and reliability of the information gathered and understand how ambiguities were resolved during the review.

Sampling and sample size

All studies meeting the inclusion criteria will be retained for analysis. As this is a scoping review, no sampling will be conducted; instead, all relevant studies identified during screening will be included. There is no minimum sample size requirement. If additional sources are needed to enhance coverage for supplementary analyses, they will be incorporated to ensure comprehensive representation of the literature.

Example of search strategy (Table 2)

This strategy employs Medical Subject Headings (MeSH) terms and title/abstract keywords to capture studies related to HIV, traditional and religious beliefs, and the African context. A multi-database search will ensure a comprehensive collection of relevant studies. To further enhance the breadth and depth of literature retrieval, we will supplement database searches with gray literature and expert consultations, addressing potential gaps. This rigorous approach strengthens the validity and reliability of our review findings, aligning with best practices for thorough reviews.

### 2.3. Stage 3: Study Selection

Inclusion criteria

Studies will be included if they meet the following criteria:Alignment with PCC framework: studies must address the Population, Concept, and Context (PCC) parameters as defined in Table 1.Study design: includes empirical research studies such as randomized controlled trials, cohort studies, case-control studies, cross-sectional studies, and qualitative studies.Language and publication date—studies must be published in English between 2004 and the present to ensure relevance to current practices and interventions.

Exclusion criteria

Studies will be excluded if they meet any of the following criteria:Focus on individual-level interventions—excludes studies focusing solely on medical treatments or individual-level interventions without considering traditional and religious beliefs or leaders.Age group exclusivity: excludes studies focusing only on populations outside adolescents and adults (e.g., children or older adults).Methodological quality: excludes studies lacking methodological rigor or insufficient data to assess the influence of traditional and religious beliefs on HIV testing and prevention.Non-empirical publications: excludes literature reviews (excluding systematic reviews), book chapters, commentaries, editorials, and opinion pieces.Descriptive or theoretical focus: exclude studies providing only descriptive data or theoretical discussions without empirical evidence.Availability of full text: excludes studies where the full text is unavailable.Language and publication date: excludes studies not in English unless a translated version is available, as well as studies published before 2004.

Study setting

This scoping review will include studies conducted across Africa, recognizing the continent’s diverse cultural, religious, and contextual factors influencing HIV testing and prevention. By incorporating research from different African settings, this review aims to provide a comprehensive understanding of the role of traditional and religious beliefs and leaders in shaping HIV-related behaviors. This inclusive approach supports the development of culturally sensitive and effective HIV interventions.

Article screening and selection process

The study selection will follow three screening stages: title screening, abstract screening, and full-text screening.

Title screening—two independent reviewers (TZ and SJN) will screen study titles based on the predefined inclusion and exclusion criteria. Discrepancies will be resolved through discussion.Abstract screening—abstracts will be screened independently by TZ and SJN following the same criteria. Disagreements will be discussed, with a third reviewer consulted if necessary.Full-text screening—selected full-text articles will be assessed independently by TZ and SJN. A third reviewer will randomly validate 10% of the selected full-text studies to ensure reliability and validity. Any conflicts will be resolved through discussion, with third-reviewer involvement as needed.

To minimize potential bias, reviewers will only see the title, abstract, keywords, and publication year, while author names, journal names, and publisher details will be blinded. This ensures that study selection is based solely on content.

All reviewers will follow a standardized screening protocol for consistency. Rayyan will be used for screening, and a librarian will assist in refining the search process. The Preferred Reporting Items for Systematic Reviews and Meta-Analyses (PRISMA) framework [17] will be used to summarize the study selection process.

### 2.4. Stage 4: Charting the Data

The data charting and extraction process is as follows:

To ensure a systematic and transparent approach, we will develop a structured data charting form tailored to our research objectives as shown in (Table 3). This form will be piloted on a small subset of studies to refine its applicability before full extraction. Reviewers will receive training to ensure standardization in data charting rather than employing blinding techniques, which are uncommon in scoping reviews.

Two independent reviewers will chart data from eligible studies. Any discrepancies will be resolved through discussion, with a third reviewer making the final decision if needed. All resolutions will be documented for transparency. The extracted data will include study characteristics, population demographics, methodology, key findings, and insights on the influence of traditional and religious beliefs in HIV testing and prevention.

Adherence to reporting standards

To enhance transparency and reproducibility, the scoping review protocol and final report will adhere to the PRISMA-ScR (Preferred Reporting Items for Systematic Reviews and Meta-Analyses extension for Scoping Reviews) checklist [25]. 

Throughout the review process, we will maintain detailed records of all decisions, including study inclusion and exclusion criteria and the methods used for data extraction. This detailed documentation will support the robustness of our findings and facilitate future updates to the review.

By following the PRISMA-ScR guidelines and ensuring thorough documentation, our aim is to produce a transparent and reproducible scoping review that effectively maps the influence of traditional and religious beliefs, and the roles of traditional and religious leaders, on HIV testing and prevention behaviors across various African contexts.

The finalized data charting form will be updated iteratively throughout the process to ensure completeness and relevance. Findings will be synthesized narratively and presented using tables and thematic groupings, aligned with PRISMA-ScR guidelines.

Data extraction and synthesis

The data extraction process for this scoping review will be conducted in several stages to ensure consistency and reliability. Initially, a training session will familiarize reviewers with the data extraction form and inclusion/exclusion criteria. This will be followed by a pilot extraction phase on a subset of included studies to refine the extraction form and ensure feasibility.

During the reliability verification stage, at least two independent reviewers will extract data from each study. A random sample of 10% will be cross-verified for discrepancies, with any disagreements resolved through discussion or a third reviewer. In the primary extraction stage, key data elements, including variables, effect sizes, and qualitative findings, will be captured. Simultaneously, a risk of bias assessment will be performed for each study using established tools. The final extraction stage will involve reviewing the extracted data for completeness and addressing any discrepancies from previous stages.

Masking will be employed to minimize bias during the extraction process. Extractors, trained research assistants, will be blinded to the research questions and hypotheses of the review, ensuring that they focus solely on extracting relevant data from the studies based on standardized coding instructions. This ensures objective data extraction without preconceived notions.

The data extraction will use a standardized charting form (Table 3) to capture study characteristics, intervention details, outcomes, and key findings. Special emphasis will be placed on identifying risk and protective factors related to HIV testing and prevention behaviors within African contexts, considering individual, family, community, and structural levels.

Temporal and policy contextualization: to account for shifts in HIV prevention policies, the following information will be extracted and analyzed:Publication year: records the year that each study was published to facilitate chronological analysis.Study period: documents the time frame during which each study was conducted to align findings with relevant policy phases.Policy context: notes any references to specific HIV prevention policies, such as the introduction of the UNAIDS 90-90-90 targets or the “Treat All” strategy.

Analysis strategy

Temporal stratification: studies will be organized based on significant policy milestones to observe trends and shifts over time.

Contextual interpretation: analysis of how changes in policy may have influenced the role of traditional and religious beliefs and leaders in HIV prevention efforts will be conducted.

To ensure reliability, the data extraction process will involve the following:Pilot testing: the form will be piloted with the review team to ensure that all necessary data are captured effectively.Independent extraction: at least two reviewers will extract data independently to enhance consistency.Discrepancy resolution: any disagreements will be resolved through discussion, with consultation of a third reviewer if needed.Reconciliation procedure: if extractors disagree on specific data points, they will engage in discussion to resolve the conflict. If consensus is not reached, a third reviewer will be consulted to make a final decision and ensure that the data are accurately extracted.

Synthesis plan

To minimize bias during the synthesis process, synthesists will be provided with neutral, unbiased instructions to ensure an objective analysis, free from preconceived notions. If multiple researchers are involved in the synthesis, they will not be informed of the research expectations or specific details during the synthesis phase to maintain impartiality. At least two independent synthesists will be involved to provide diverse perspectives and reduce individual bias. If discrepancies arise in the synthesis, a third synthesist will be consulted to resolve any differences.

Once data extraction is complete, the information will undergo thematic analysis to identify patterns and relationships across the studies. This will involve coding the data manually or using software tools, depending on the volume, and organizing the findings into key themes related to traditional and religious beliefs and their impact on HIV testing and prevention behaviors. The synthesis will be conducted in two stages:Primary analysis: a narrative synthesis of the findings will be undertaken, drawing connections between the themes identified during the extraction.Secondary analysis: if applicable, subgroup analyses will be conducted to explore variations across different studies, settings, or populations. Special attention will be paid to studies conducted in African contexts, ensuring that cultural considerations are integrated into the analysis.

This synthesis aims to map the existing literature, summarize key themes, and identify gaps rather than offering a detailed analysis of outcomes. This process will enable a comprehensive synthesis of the literature, highlighting key themes, identifying gaps, and mapping the available evidence relevant to the research questions.

The criteria for conclusions/inference criteria are as follows.

In drawing conclusions from the synthesis of the extracted data, pre-specified criteria will be applied. These criteria include a minimal effect size or level of significance for the relationships between traditional and religious beliefs and HIV testing behaviors, as well as the presence of consistent patterns across studies. If themes are insufficiently represented or findings are inconsistent, conclusions will be drawn with caution, acknowledging the limitations. Subgroup analyses, if applicable, will focus on variations across different populations or settings, with particular attention to studies conducted within African contexts to ensure relevance and cultural alignment.

Handling missing data

For missing data, every effort will be made to retrieve missing entities by contacting study authors for clarification or additional information. If the authors do not respond or cannot provide the missing data, the following steps will be taken: for quantitative data, missing effect sizes or statistical values will be imputed using available related data, and, if imputation is not possible, the study will be excluded from the analysis. For missing qualitative data, studies will be included in the synthesis with available information, and any limitations due to missing data will be clearly acknowledged. If substantial data are missing and cannot be retrieved, the study will be excluded, and this will be noted in the final documentation. This approach ensures a structured method to handle missing data while minimizing potential bias.

Data validation

To ensure the accuracy and usefulness of the extracted data, several steps will be taken during the validation process. First, we will conduct a thorough review of the extracted data for consistency and completeness. Any outliers or inconsistencies in the data will be identified and flagged for further investigation. This includes identifying any discrepancies between the reported data and what is presented in the source material, such as errors in effect sizes or methodological inconsistencies.

If any studies are found to have been retracted or corrected post-publication, they will be excluded from the review to maintain the validity of the analysis. Additionally, triangulation with other sources will be employed to cross-check the findings. For example, if certain data points are ambiguous or incomplete, alternative reliable sources (such as secondary data or related studies) will be consulted to corroborate the findings.

Data validity will be assessed based on predefined criteria, such as the methodological rigor of the study, the clarity and transparency of data reporting, and the relevance to the research questions. Any data that violate these criteria, such as poor reporting quality or studies with methodological flaws that compromise the integrity of the findings, will be excluded from the analysis. Studies found to be invalid or that contain significant biases will be noted, and their exclusion will be justified in the review findings.

### 2.5. Stage 5: Collating, Summarizing, and Reporting the Results

In this scoping review, we will systematically collate, summarize, and report findings on the influence of traditional and religious beliefs, as well as the roles of traditional and religious leaders, on HIV testing and prevention behaviors across diverse African contexts.

Study selection presentation

The study selection process will be transparently depicted using a PRISMA-ScR flowchart, detailing the number of records identified, included, and excluded, and explaining reasons for exclusions.

Data extraction

Extracted data will include the following key information: author(s) names, year of publication, study design, study population, intervention(s), study setting, aims, geographic location, and outcomes. This will provide a structured overview of the existing evidence on HIV testing and prevention behaviors within the African context.

Data analysis and synthesis

A narrative synthesis approach will be employed to analyze the extracted data. Findings will be organized thematically and classified according to the types of interventions and their effectiveness. Risk and protective factors associated with HIV testing and prevention behaviors will be categorized at individual, family, community, and structural levels. An inductive categorization approach will be used to capture emerging factors from the data.

Reporting findings

Findings will be presented using tables, charts, maps, conceptual frameworks, and narrative summaries, allowing for clear interpretation and cross-regional comparisons. Conceptual mapping will be used to visually represent relationships between intervention types, risk and protective factors, and contextual influences [21].

Implications

This review will discuss the implications of the findings for future research, practice, and policy formulation, aiming to provide a basis for evidence-informed decision making in HIV testing and prevention strategies. It will highlight gaps in the existing literature and suggest areas for further research.

Quality considerations

In line with the PRISMA-ScR guidelines, this scoping review will not perform a formal quality assessment of the included studies, as the primary aim is to map the existing literature and identify key concepts, gaps, and types of evidence related to traditional and religious beliefs influencing HIV testing and prevention behaviors in African contexts. While methodological rigor and trustworthiness will not be formally assessed, we will consider the context and design of each study during data synthesis and interpretation to provide a comprehensive and contextually relevant analysis. By adhering to the PRISMA-ScR guidelines and employing a systematic approach to data extraction and synthesis, this review aims to offer valuable insights into how traditional and religious beliefs, as well as the roles of traditional and religious leaders, influence HIV testing and prevention behaviors across diverse African settings.

### 2.6. Stage 6: Stakeholder Consultation

Stakeholder consultation is a critical aspect of Arksey and O’Malley’s scoping review framework [13], ensuring that findings are relevant, comprehensive, and applicable to real-world contexts. In this scoping review, stakeholder engagement will play a pivotal role in refining results and enhancing their practical utility.

Identification of key stakeholders

Stakeholders will be selected based on their expertise and involvement in areas pertinent to the study. Key stakeholders will include the following:Traditional healers—recognized for their influence in community health practices and their role in HIV prevention and care.Religious leaders—their perspectives are crucial, given the impact of religious beliefs on health behaviors and HIV-related stigma.Public health practitioners—professionals engaged in implementing HIV prevention strategies and interventions.Policymakers—individuals involved in shaping health policies that affect HIV prevention and care services.Researchers—scholars with expertise in HIV, traditional medicine, religious studies, or African health systems.Representatives from organizations—organizations focused on adolescent health, sexual and reproductive health, and HIV and AIDS advocacy.

Consultation methods

Depending on availability and logistical considerations, consultations will be conducted through the following:Virtual meetings—utilizing digital platforms to facilitate discussions across diverse geographic locations, ensuring inclusivity and flexibility.In-person meetings—organizing face-to-face engagements where feasible to foster deeper dialogue and relationship building, enhancing trust and collaboration.Structured discussions—facilitated conversations focusing on specific topics to gather targeted insights, promoting in-depth understanding.Surveys and focus groups—employing these tools to collect quantitative and qualitative data on stakeholder perspectives, capturing a broad range of opinions and experiences.

Integration of stakeholder feedback

Stakeholder input will be systematically integrated into the review process:Validation of results—ensuring that findings accurately reflect the realities and perspectives of those directly involved in HIV prevention and care.Identification of gaps—highlighting areas where additional research or intervention is needed, based on stakeholder experiences and observations.Refinement of key themes—adjusting and clarifying themes to better align with practical applications and cultural contexts.

This consultation will occur during the final stages of data synthesis, allowing stakeholders to inform the interpretation of results and help to shape actionable recommendations. By incorporating diverse perspectives, this review aims to produce findings that are not only academically rigorous but also culturally relevant and practically applicable in enhancing HIV prevention strategies across African communities.

Dissemination plan

Findings will be submitted to journals specializing in public health, HIV prevention, and cultural practices, with a specific focus on studies related to Africa.The results will be presented at both national and international conferences focused on HIV prevention, public health, and community health practices in Africa. These platforms will encourage collaboration, knowledge exchange, and further discussions on the role of traditional and religious beliefs in shaping HIV testing and prevention behaviors.Actionable policy briefs will be prepared for health policymakers, community leaders, religious and traditional healers, and public health officials. These briefs will provide recommendations on integrating traditional and religious beliefs into HIV prevention strategies and inform culturally sensitive health interventions.This study’s data, including key findings, will be shared publicly on the Open Science Framework (OSF) to promote transparency, facilitate reproducibility, and support further research on HIV prevention behaviors within African contexts.

These dissemination strategies are designed to ensure that the review’s findings reach key stakeholders, facilitating evidence-based decision making and contributing to the development of more effective, culturally appropriate HIV testing and prevention strategies in Africa.

### 2.7. Study Limitations

This scoping review acknowledges several limitations:Language bias—limiting the review to English language publications may exclude relevant studies published in other languages.Publication bias—relying on publicly available data may exclude unpublished or gray literature.Methodological diversity—including studies with varying methodologies and quality may affect the reliability of the synthesized findings.Contextual interpretation—interpreting traditional and religious beliefs across diverse African contexts is complex and could result in the oversimplification or misinterpretation of nuanced cultural practices.Evolving beliefs—beliefs and practices related to HIV testing and prevention may have evolved over time, and the review might not fully capture current perspectives.

## 3. Discussion

This scoping review protocol outlines our plan to systematically examine how traditional and religious beliefs, and the roles of traditional and religious leaders, influence HIV testing and prevention behaviors across diverse African contexts. These cultural and religious factors are crucial for developing public health initiatives that are both culturally sensitive and effective. Despite significant advancements in HIV prevention, cultural and religious beliefs continue to profoundly influence healthcare behaviors in Africa. Faith-based organizations (FBOs) play a pivotal role in HIV prevention efforts across the continent.

Guided by Arksey and O’Malley’s six-stage framework, we will begin by identifying the research question, followed by comprehensive searches for relevant studies, rigorous selection processes, and detailed data extraction using standardized forms. A key aspect of this process is stakeholder consultation, which will be integrated early into the methodology to ensure that findings are relevant and practically applicable. Engaging traditional healers, religious leaders, public health practitioners, policymakers, researchers, and organizational representatives during the data synthesis phase will validate and refine interpretations. This ensures that the review not only offers an academic contribution but also provides actionable insights grounded in real-world perspectives.

By addressing the gap in the existing literature on the influence of traditional and religious beliefs on HIV testing and prevention behaviors, this review will contribute new insights into how these cultural factors shape health interventions in African contexts. The review will also highlight specific regions where such factors have not been explored adequately, offering new directions for research and practice.

Subsequently, we will analyze and synthesize the data thematically and consult with key stakeholders—including traditional healers, religious leaders, public health practitioners, policymakers, researchers, and organizational representatives—to validate findings and refine our interpretations. By integrating cultural and religious perspectives, this review aims to inform the development of effective and culturally appropriate HIV prevention programs across African communities.

## Figures and Tables

**Table 1 ijerph-22-00743-t001:** Population concept and context.

Population	Individuals and communities in African countries
Concept	The influence of traditional and religious beliefs on attitude and behaviors toward HIV testing and prevention.
Context	African communities where traditional and religious beliefs significantly impact health behaviors.

**Table 2 ijerph-22-00743-t002:** Example search strategy for PubMed.

(“HIV” [Mesh] OR “HIV Infections” [Mesh] OR “acquired immunodeficiency syndrome” [Mesh] OR “HIV” OR “AIDS” AND (“Religion and Medicine” [Mesh] OR “Religion” [Mesh] OR “Spirituality” [Mesh] OR “Faith-Based Organizations” [Mesh] OR “Religious Beliefs” OR “Traditional Beliefs” OR “Spiritual Beliefs” AND (“Africa” [Mesh] OR “Africa South of the Sahara” [Mesh] OR “Sub-Saharan Africa” [Title/Abstract] OR “African Countries” [Title/Abstract])

**Table 3 ijerph-22-00743-t003:** Data charting form.

Study ID—unique identifier (author name, tittle, year of publication). Study design—type of study (qualitative, quantitative, mixed methods). Setting—country/region of study. Population—sample characteristics (age, gender, religious/traditional affiliation, etc.). Data source—published literature, gray literature, or other sources. Key findings—main results related to the study objective. Influence of traditional and religious beliefs—factors related to traditional and religious beliefs shaping HIV testing and prevention behaviors

## Data Availability

No new data were created or analyzed in this study. This article presents a scoping review protocol and does not involve the generation of primary data. Upon completion of the scoping review, all datasets generated or extracted will be made openly available through the Open Science Framework (OSF) repository, ensuring full transparency and accessibility for future use and validation.

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
