# Peer review of "The Role of Traditional and Religious Beliefs in HIV Testing and Prevention in Africa: A Scoping Review Protocol"

_ijerph, 2025, doi:10.3390/ijerph22050743_

Round 1

Reviewer 1 Report

Comments and Suggestions for Authors

The role of traditional and religious beliefs in HIV testing and prevention in Africa: a scoping review protocol

Introduction

While biomedical approaches have been extensively studied and implemented, the influence of traditional and religious beliefs on attitudes and behaviors related to HIV testing and prevention has not received equivalent attention [2, 3]. Understanding these cultural and religious contexts is essential for developing interventions that are both culturally sensitive and effective [4]. It is important to provide concrete/tangible examples of traditional and religious beliefs on attitudes and bahaviors related to HIV testing and prevention. And also, to provide information on why […] HIV testing and prevention has not received equivalent attention […]

In many African societies, cultural and religious frameworks significantly influence health-seeking behaviors. Faith-based organizations (FBOs) and religious leaders often play pivotal roles in community life, shaping perceptions and responses to health interventions (how? Provide concrete details on that).

Studies have shown that FBOs are accessible and influential in delivering HIV prevention messages, making them important players in HIV prevention efforts [6]. In sub-Saharan Africa for example, FBOs have been at the forefront of responding to the HIV and AIDS epidemic, delivering affordable and acceptable health promotion messages [6, 7], thereby reducing stigma and increasing community engagement. Training these leaders with accurate HIV information has been shown to enhance their effectiveness in disseminating knowledge and supporting affected individuals within their congregations [8] (you must provide details on the type of stigma mentioned and trace it to specific geographic space)

However, certain traditional and religious beliefs can also pose challenges to HIV testing and prevention (what are some of the traditional and religious beliefs that can pose challenges to HIV testing and prevention)? Religious beliefs can both positively and negatively influence adherence to HIV treatment regimens, highlighting the need for culturally sensitive interventions. For instance, while some beliefs may facilitate supportive environments that enhance adherence, others contribute to fatalistic attitudes that hinder participation in treatment [9] (provide concrete details on these beliefs to substantiate your argument). Barriers such as stigma (example)? cultural shame (what is it)? and misconceptions (example)? about HIV transmission persist in some communities, deterring individuals from seeking testing or preventive measures [10]. Additionally, beliefs in faith healing and prayers as substitutes for medical treatments have been identified as factors that may discourage HIV testing and treatment adherences [11, 12] (did you find out why beliefs in faith healing and prayers are substitutes for medical treatments discourage HIV testing and treatment adherences)?

Rationale of the study

This scoping review aims to synthesize existing research on how traditional and religious beliefs influence attitudes and behaviors toward HIV testing and prevention in Africa. By mapping the current literature, identifying knowledge gaps, and analyzing the roles of religious and traditional leaders, this study will provide valuable insights to inform the development of culturally sensitive strategies that align with local beliefs (what is local beliefs? How does behaviours and attitudes become local beliefs)? and practices. The findings will assist policymakers, public health professionals, and researchers in designing more effective HIV interventions across diverse African communities (the fact that you acknowledge that African communities are diverse, means the behaviours and attitudes may be diverse or similar)? This study would have been best placed in one specific regional economic community). In this case, then you may proceed to do data triangulation in other to find out whether the claims of this study are the same across the continent which you emphasize has diverse communities).

Research questions and research objectives

  1. To systematically map existing literature on the influence of traditional and religious beliefs on HIV testing and prevention in Africa (remember research objectives are the flip side of research questions. Your objective 1 is blur when looking at research question 1. There may be a need to fine-tune the objective or the question. Or you could say; to establish the influence of traditional and religious beliefs on HIV testing and prevention in Africa […])

Sampling and sample size

All studies meeting the inclusion criteria (what is this inclusion criteria? Is it a sampling technique)? will be retained for analysis. As this is a scoping review, no sampling will be conducted; instead, all relevant studies identified during screening will be included. There is no minimum sample size requirement (It is true there is no universally "zero" minimum sample size. However, the necessity of a minimum sample size depends heavily on the specific research question, statistical analysis, and the type of study. Some studies, such as, qualitative research or case studies, may require a smaller sample size than others). Reading through your materials and methods, I suggest there is need justify why there is no minimum sampling size requirement).

If additional sources are needed to enhance coverage for supplementary analyses, they will be incorporated to ensure comprehensive representation of the literature (how would there be incorporated).

Make sure to add a few lines on validity (whether a study measures what it intends to measure), and reliability (indicates consistency and stability of the results across different times, samples, or methods. Remember, validity and reliability are crucial for producing high-quality, trustworthy research findings).

Author Response

Comment 1: It is important to provide concrete/tangible examples of traditional and religious beliefs on attitudes and behaviours related to HIV testing and prevention. 

Response 1: Thank you for your insightful feedback regarding the need to provide concrete examples of traditional and religious beliefs influencing attitudes and behaviours related to HIV testing and prevention. We acknowledge the importance of illustrating these influences to underscore the relevance and depth of our study.​ In our revised introduction, we have incorporated specific examples to elucidate how traditional and religious beliefs can both facilitate and hinder HIV-related health behaviours. We have updated the introduction to reflect these examples, thereby providing a more comprehensive context for our study. This enhancement aims to clarify the significance of our research objectives and the rationale behind exploring the influence of traditional and religious beliefs on HIV testing and prevention in Africa. Page 2, line 46 - 78. 

Comment 2: And also, to provide information on why […] HIV testing and prevention has not received equivalent attention

Response 2: Thank you for your comment. We have revised our introduction and we have incorporated these the reasons. Page 3, line 81 -96. 

Comment 3: In many African societies, cultural and religious frameworks significantly influence health-seeking behaviors. Faith-based organizations (FBOs) and religious leaders often play pivotal roles in community life, shaping perceptions and responses to health interventions (how? Provide concrete details on that).

Response 3: Thank you for your comment. We revised and added this information in the manuscript. Page 1- 2, line 30 -42.

Comment 4: Thereby reducing stigma and increasing community engagement. Training these leaders with accurate HIV information has been shown to enhance their effectiveness in disseminating knowledge and supporting affected individuals within their congregations [8] (you must provide details on the type of stigma mentioned and trace it to specific geographic space).

Response 4: Thank you for the comment. We have included an example of the type of the stigma. Page 2, line 63 -64.

Comment 5: However, certain traditional and religious beliefs can also pose challenges to HIV testing and prevention (what are some of the traditional and religious beliefs that can pose challenges to HIV testing and prevention)?

Response 5: Thank you for the comment. We have incorporated some challenges that can be posed by traditional and religious beliefs that can pose challenges to HIV testing and prevention. Page 2, line 68 - 78. 

Comment  6: Religious beliefs can both positively and negatively influence adherence to HIV treatment regimens, highlighting the need for culturally sensitive interventions. For instance, while some beliefs may facilitate supportive environments that enhance adherence, others contribute to fatalistic attitudes that hinder participation in treatment [9] (provide concrete details on these beliefs to substantiate your argument).

Response 6: ​Thank you for your question regarding the integration of specific traditional and religious beliefs that may hinder HIV testing and prevention. In response to the reviewer's comment, we have revised the introduction to include concrete examples of such beliefs, supported by relevant literature. Page 2, Line 47 -60

Comment 7: barriers such as stigma (example), cultural shame (what is it)? and misconceptions (example)?

Response 7: Thank you for your comments. We have revised the introduction and we have lost some of the statements as they no longer appear in our introduction. Some of the information is covered on Page 2, line 47 -60.

Comment 8: Additionally, beliefs in faith healing and prayers as substitutes for medical treatments have been identified as factors that may discourage HIV testing and treatment adherences [11, 12] (did you find out why beliefs in faith healing and prayers are substitutes for medical treatments discourage HIV testing and treatment adherences)?

Response 8: Thank you for your comment. We have revised our introduction and we have incorporated  specific examples and explanations of how certain beliefs in faith healing and prayer may discourage individuals from seeking HIV testing and adhering to antiretroviral therapy (ART). Pages 2 - 3, line 69 - 78.

Comment 9:  Rationale of the study: By mapping the current literature, identifying knowledge gaps, and analysing the roles of religious and traditional leaders, this study will provide valuable insights to inform the development of culturally sensitive strategies that align with local beliefs (what is local beliefs? How does behaviours and attitudes become local beliefs)? and practices. The findings will assist policymakers, public health professionals, and researchers in designing more effective HIV interventions across diverse African communities (the fact that you acknowledge that African communities are diverse, means the behaviours and attitudes may be diverse or similar)? This study would have been best placed in one specific regional economic community). In this case, then you may proceed to do data triangulation in other to find out whether the claims of this study are the same across the continent which you emphasize has diverse communities).

Response 9: Thank you for your insightful feedback on the rationale section of our study. We appreciate the opportunity to clarify and elaborate on the points you've raised.​ We revised our rationale as on Page 3, line 98 -103. 

Clarification on Local Beliefs: In our study, 'local beliefs' refer to the collective understandings, values, and practices that are specific to particular communities or cultural groups within Africa. These beliefs are shaped by a confluence of historical experiences, religious teachings, traditional customs, and social interactions. They influence how individuals perceive health, illness, and appropriate health-seeking behaviors. For instance, in some communities, illnesses like HIV are interpreted through spiritual lenses, such as being a punishment from ancestral spirits or a result of witchcraft. These interpretations can significantly impact individuals' decisions to seek testing or adhere to treatment regimens.

Addressing the Diversity of African Communities: We acknowledge that African communities are diverse, with varying cultural, religious, and social contexts. This diversity means that behaviors and attitudes towards HIV prevention and treatment can differ significantly across regions. While some beliefs and practices may be prevalent across multiple regions due to shared cultural or religious influences, variations do exist, and interventions must be tailored to address specific community contexts. Our study aims to capture this diversity by systematically mapping existing literature across different African contexts, thereby identifying both commonalities and unique challenges in HIV prevention efforts.

Consideration of regional focus and data triangulation: We appreciate your suggestion regarding focusing the study on a specific regional economic community (REC) to provide more detailed insights and allow for more targeted recommendations. While our current scoping review aims to provide a broad overview of the influence of traditional and religious beliefs on HIV prevention across Africa, we recognize the value of conducting more region-specific studies. Such studies could employ data triangulation methods to assess the applicability of our findings within specific regions, ensuring a comprehensive understanding of the continent's diverse cultural landscapes. We will consider this approach in future research endeavors to enhance the specificity and applicability of our findings.​

We have revised the rationale section of our manuscript to incorporate these clarifications and to better reflect the complexity and diversity of local beliefs and practices across African communities. We believe these revisions will strengthen the study's foundation and provide clearer guidance for policymakers, public health professionals, and researchers in designing culturally sensitive HIV interventions. page 2, line 52 -53.

Comment 10: Research questions and research objectives: To systematically map existing literature on the influence of traditional and religious beliefs on HIV testing and prevention in Africa (remember research objectives are the flip side of research questions. Your objective 1 is blur when looking at research question 1. There may be a need to fine-tune the objective or the question. Or you could say; to establish the influence of traditional and religious beliefs on HIV testing and prevention in Africa […])

Response 10:  Thank you for your comment. We have revised our research questions and the objectives. Page 5 - 6, line 179 -195. 

Comment 11: All studies meeting the inclusion criteria (what is this inclusion criteria? Is it a sampling technique)? will be retained for analysis. As this is a scoping review, no sampling will be conducted; instead, all relevant studies identified during screening will be included. There is no minimum sample size requirement (It is true there is no universally "zero" minimum sample size. However, the necessity of a minimum sample size depends heavily on the specific research question, statistical analysis, and the type of study. Some studies, such as, qualitative research or case studies, may require a smaller sample size than others). Reading through your materials and methods, I suggest there is need justify why there is no minimum sampling size requirement).

Response 11: Thank you for your comment. In the context of this scoping review, the inclusion criteria refer to the predefined parameters that determine which studies are eligible for inclusion. These criteria encompass aspects such as the population studied, the concepts explored, the context of the research, and the types of evidence considered. This approach aligns with the methodological guidance provided by the Joanna Briggs Institute (JBI), which emphasizes the importance of clearly defined inclusion criteria to ensure the relevance and comprehensiveness of scoping reviews sampling techniques. Instead, they aim to include all relevant literature that meets the inclusion criteria, thereby providing a comprehensive overview of the existing evidence on a particular topic. This methodology is designed to map the breadth and depth of research in a given area, identify gaps in the literature, and inform future research directions.​

Justification for absence of minimum sample size: In this scoping review, we do not impose a minimum sample size requirement because our objective is to comprehensively map all relevant literature on the influence of traditional and religious beliefs on HIV testing and prevention in Africa. This approach aligns with the Joanna Briggs Institute's guidance for scoping reviews, which emphasizes the inclusion of all pertinent studies regardless of sample size to capture the breadth of existing evidence. ​By including studies of varying sample sizes, we aim to identify patterns, themes, and gaps in the literature, providing a robust foundation for future research and policy development. This inclusive strategy ensures that valuable insights are not overlooked due to arbitrary sample size thresholds. Unlike primary research studies that involve sampling participants, scoping reviews do not employ traditional. 

Comment 12: If additional sources are needed to enhance coverage for supplementary analyses, they will be incorporated to ensure comprehensive representation of the literature (how would there be incorporated).

Response 12: Thank you for your comment. In this scoping review, we aim to provide a comprehensive overview of the literature on the influence of traditional and religious beliefs on HIV testing and prevention in Africa. To ensure a thorough representation, we will include all studies that meet our predefined inclusion criteria, encompassing various methodologies and contexts. Should additional sources be identified during the review process, such as through citation tracking, expert consultations, or grey literature searches, they will be incorporated systematically. Each new source will undergo the same rigorous screening and data extraction procedures as the initially identified studies to maintain consistency and transparency. This approach aligns with best practices in scoping reviews, which emphasize the importance of capturing a broad spectrum of relevant evidence to inform comprehensive analyses and identify research gaps.

Comment 13: Make sure to add a few lines on validity (whether a study measures what it intends to measure), and reliability (indicates consistency and stability of the results across different times, samples, or methods. Remember, validity and reliability are crucial for producing high-quality, trustworthy research findings).

Response 13: Thank you for your comment. In this scoping review, to ensure a thorough representation, we will include all  studies that meet our predefined inclusion criteria, encompassing various methodologies and contexts. Should additional sources be identified during the review process, such as through citation tracking, expert consultations, or grey literature searches—they will be incorporated systematically. Each new source will undergo the same rigorous screening and data extraction procedures as the initially identified studies to maintain consistency and transparency. This approach aligns with best practices in scoping reviews, which emphasize the importance of capturing a broad spectrum of relevant evidence to inform comprehensive analyses and identify research gaps. All this information is covered in different areas in the manuscript, 

Page 7, line 242, line 266 -267, line 283, 

Page 8, line 328 -330.

Page 10, line 409

Page 11, line 470 -471.

Page 11, line 475 - 477

Reviewer 2 Report

Comments and Suggestions for Authors

Excellent proposal! Clearly stated, rigorous, empirically based study on a topic of high importance.

Provide a  bit more information on each of the 10 data bases in an appendix.  Do they overlap?  What are the limitations of each? 

Perhaps should conduct the research in multiple stages.  Why not just stick to the 10 data bases; present preliminary findings;, and then -- if results are inconclusive -- move on to additional search strategies?

Discussions of preliminary findings with religious leaders, policy makers, and traditional healers may end up being the major contribution.     

Author Response

Comment 1: Provide a  bit more information on each of the 10 data bases in an appendix.  Do they overlap?  What are the limitations of each?

Response 1: Thank you for this comment. We have compiled an appendix detailing the ten databases selected for our literature search. Appendix 1. This includes information on each database's scope, strengths, limitations, and potential overlaps with other databases.

Comment 2: Perhaps should conduct the research in multiple stages.  Why not just stick to the 10 data bases; present preliminary findings;, and then -- if results are inconclusive -- move on to additional search strategies. 

Response 2: Thank you for your insightful suggestion. In alignment with best practices for scoping reviews, we have adopted a multi-stage search strategy to ensure a comprehensive and systematic identification of relevant literature. Initially, we will conduct a thorough search across the ten selected databases—PubMed, Scopus, Web of Science, AJOL, Africa-Wide Information, CINAHL, PsycINFO, ERIC, LLBA, and Embase—to capture a broad spectrum of studies pertinent to our research questions. This approach is consistent with the JBI's recommended three-step search process, which includes an initial limited search, analysis of keywords and index terms, and a subsequent comprehensive search across all chosen databases . Should the initial search yield inconclusive or insufficient results, we are prepared to implement additional search strategies. These may involve exploring grey literature sources, hand-searching key journals, and examining the reference lists of included studies to identify further relevant publications. This iterative approach allows for flexibility and responsiveness to the findings of the preliminary search, ensuring that our review is both exhaustive and focused.​ By structuring our search in multiple stages, we aim to balance comprehensiveness with efficiency, minimizing the risk of overlooking pertinent studies while managing resources effectively. This methodology enhances the validity and reliability of our scoping review, providing a robust foundation for mapping existing literature and identifying knowledge gaps in the influence of traditional and religious beliefs on HIV testing and prevention in Africa. This is covered in Page 6, line 199 -219. 

Comment 3: Discussions of preliminary findings with religious leaders, policy makers, and traditional healers may end up being the major contribution.     

Response 3: Thank you for your insightful observation regarding the potential significance of engaging religious leaders, policymakers, and traditional healers in discussions of preliminary findings. We concur that such engagements can substantially enhance the impact and applicability of our study. 

By integrating discussions with religious leaders, policymakers, and traditional healers into our research process, we aim to ensure that our findings are not only academically robust but also practically relevant and culturally sensitive. We believe this approach will enhance the study's contribution to effective HIV interventions across diverse African communities.​

We appreciate your valuable feedback, which has contributed to strengthening the relevance and impact of our study.

Reviewer 3 Report

Comments and Suggestions for Authors

Dear Authors,

Congratulations on undertaking this important study exploring the relevance of faith-based organizations (FBOs) and their leaders in HIV prevention. This is a timely and significant topic, and I commend your initiative. Below are some detailed suggestions to improve the clarity, coherence, and rigor of your protocol manuscript:

Abstract

  • Consider reducing the use of emphatic adverbs and adjectives such as “significant,” “profoundly,” and “comprehensive.” The message remains strong without these qualifiers and aligns better with the current stage of the study (protocol development).

Introduction

  • Line 30: The UNAIDS report cited is from 2021. Please consider using the most recent report, as these are updated annually and could provide more current data.

  • Line 34: Minor formatting issue — the first paragraph ends without a period.

  • Lines 37–40: You suggest that cultural competence improves treatment adherence. Please support this with an appropriate citation.

  • Lines 50–51: These two lines convey similar messages. I recommend removing the earlier sentence to avoid redundancy.

  • Lines 63–65: The phrase “some religions” may be problematic unless backed by specific literature. If unsupported, consider rephrasing as “some groups” or “certain communities.”

  • In the final paragraph, you mention that the study aims to assess the African context. Please clarify whether this includes the entire African continent or is restricted to sub-Saharan Africa, as used elsewhere in the manuscript. This distinction is critical for understanding the study’s scope.

General suggestion: I suggest restructuring the Introduction into three clear paragraphs for coherence:

  1. What is known: Impact of FBOs and cultural beliefs on HIV prevention in Africa.

  2. What is lacking: Gaps and challenges in the current literature.

  3. What this study adds: The rationale and aims of your study.

Materials and Methods

  • Framework: It would strengthen your protocol to cite studies that have applied Arksey and O’Malley’s framework in the context of HIV or similar public health issues.

  • Line 85: Again, clarify whether your review covers all of Africa or is focused on sub-Saharan Africa.

  • Line 90: Please list the databases explicitly, even if a full strategy is presented later. This allows easier understanding up front.

  • Lines 95–98: You mention policy shifts in HIV prevention. Will your analysis be stratified by time period or policy phase? Clarify how you plan to ensure findings reflect such shifts.

  • Line 133: The two research questions seem quite similar. If the intent is to differentiate between individual-level beliefs and community-level leadership roles, please state this explicitly.

  • Lines 320–326: Consider removing or condensing this explanation, as it duplicates content from the previous sentence.

  • Lines 334–337: Referencing the PRISMA checklist is sufficient; this detailed description may not be necessary here.

  • Line 420 onward: The inclusion of sections on data transformations, missing data handling, and validation seems more aligned with quantitative meta-analysis than a scoping review. Please clarify their relevance in this context or consider omitting them.

  • Line 506: There appears to be an inconsistency regarding quality assessment. Earlier, you mentioned quality considerations, but here it suggests there may be none. Please clarify your approach and consolidate all quality-related discussions into one section for clarity.

  • Line 516: Stakeholder consultation is a key part of your approach. Consider referencing this earlier in the methods section.

Additional Suggestions

  • Consider reducing the overall use of emphatic adverbs and adjectives throughout the manuscript to maintain a more scientific tone.

  • If it does not restrict the scope, consider specifying “sub-Saharan Africa” in the title for clarity.

  • Line 638: Begin the sentence with a capital letter.

  • Line 643: An isolated quotation mark appears here — please remove.

Table A2: Please clarify if your search strategy will include all African countries listed individually. The current strategy may inadvertently exclude relevant studies from specific countries.

Author Response

Comment 1: Consider reducing the use of emphatic adverbs and adjectives such as “significant,” “profoundly,” and “comprehensive.” The message remains strong without these qualifiers and aligns better with the current stage of the study (protocol development).

Response 1: Thank you for your comment. We have revised the abstract accordingly and we have removed terms such as “significant,” “profoundly,” and “comprehensive” while preserving the intent and focus of the study.

Comment 2: the  Introduction: Line 30: The UNAIDS report cited is from 2021. Please consider using the most recent report, as these are updated annually and could provide more current data.

Response 2: Thank you for your comment. We have restructured our introduction, and we have removed this reference. 

Comment 3: Minor formatting issue- the first paragraph ends without a period. 

Response 3: Thank your for your comment. We have revised the manuscript and have formatted formatting issues. 

Comment 4:  Lines 37–40: You suggest that cultural competence improves treatment adherence. Please support this with an appropriate citation.

Response 4: Thank you for your comment. Upon review, we have removed the sentence suggesting that cultural competence improves treatment adherence from the introduction. This decision was made to maintain the focus of the introduction on the influence of traditional and religious beliefs on HIV testing and prevention behaviours, aligning with the primary objectives of our scoping review. We appreciate your guidance in refining the scope and clarity of our manuscript.

Comment 5: Lines 50–51: These two lines convey similar messages. I recommend removing the earlier sentence to avoid redundancy. 

Response 6: Thank you for your valuable comment. We have revised the introduction to address the redundancy identified in lines 50–51. Specifically, we have removed the earlier sentence to eliminate repetition and enhance clarity. This adjustment ensures a more concise and focused presentation of the information. We appreciate your guidance in improving the manuscript's quality. 

Comment 6: Lines 63–65: The phrase “some religions” may be problematic unless backed by specific literature. If unsupported, consider rephrasing as “some groups” or “certain communities.”

Response 7:  Thank you for your comment. We have revised the introduction and the sentence has been removed. 

Comment 7: In the final paragraph, you mention that the study aims to assess the African context. Please clarify whether this includes the entire African continent or is restricted to sub-Saharan Africa, as used elsewhere in the manuscript. This distinction is critical for understanding the study’s scope.

Response 8: Thank you for your comment. We have clarified that the study focuses on Africa. Page 4, line 113-114.

Comment 8:   General suggestion: I suggest restructuring the Introduction into three clear paragraphs for coherence: 

What is known: Impact of FBOs and cultural beliefs on HIV prevention in Africa. 

What is lacking: Gaps and challenges in the current literature. 

What this study adds: The rationale and aims of your study

Response 8. Thank you for your valuable feedback. We have restructured the Introduction into three distinct paragraphs to enhance coherence:

  1. What is known: We discuss the impact of faith-based organizations and cultural beliefs on HIV prevention in Africa.​ Page 1 -3, line 28 - 79.

  2. What is lacking: We identify gaps and challenges in the current literature, highlighting the need for further research.​ Page 3, line 83 - 98. 

  3. What this study adds: We present the rationale and aims of our study, outlining how this scoping review will contribute to the existing body of knowledge. Page 3, line 99-116.

Comment 9:  Materials and Methods Framework: It would strengthen your protocol to cite studies that have applied Arksey and O’Malley’s framework in the context of HIV or similar public health issues.

Response 9: Thank you for your comment. We have inserted a reference of a study that has used the same framework in the HIV context, Page 4, line 125 -126.

Comment 10: Line 85: Again, clarify whether your review covers all of Africa or is focused on sub-Saharan Africa.

Response 10: Thank you for the comment. We have clarified that the review will focus on Africa. Page 4, line 113. 

Comment 11: Line 90: Please list the databases explicitly, even if a full strategy is presented later. This allows easier understanding up front.

Response 11: Thank you for your comment. We have added the list of the databases that the review will use. This is on page 4, line 134-136. 

Comment 12: Lines 95–98: You mention policy shifts in HIV prevention. Will your analysis be stratified by time period or policy phase? Clarify how you plan to ensure findings reflect such shifts. 

Response 12:  Thank you for your comment. We acknowledge the importance of contextualizing our findings within the evolving landscape of HIV prevention policies. While our primary objective is to map existing literature on the influence of traditional and religious beliefs on HIV testing and prevention in Africa, we recognize that policy shifts can significantly impact these dynamics. To address this, we will:

 During the data extraction phase, we will record the publication year of each study and note any specific policy frameworks or initiatives referenced.​

We will organize the findings chronologically to observe trends and shifts over time, particularly in relation to major policy changes such as the introduction of the UNAIDS 90-90-90 targets in 2014 and the adoption of the "Treat All" strategy in 2015.​

In our analysis, we will consider how these policy shifts may have influenced the role of traditional and religious beliefs and leaders in HIV prevention efforts. This information is on page 10, line 398 - 410. 

Comment 13: Line 133: The two research questions seem quite similar. If the intent is to differentiate between individual-level beliefs and community-level leadership roles, please state this explicitly. 

Response 13: Thank you for your comment. To clarify the distinction between individual-level beliefs and community-level leadership roles, we have revised our research questions. Page 5, line 181 - 185.

Comment 14: Lines 320–326: Consider removing or condensing this explanation, as it duplicates content from the previous sentence.

Response 14: Thank you for your comment. We have removed the sentences from the protocol.  

Comment 15: Lines 334–337: Referencing the PRISMA checklist is sufficient; this detailed description may not be necessary here.

 Response 15: Thank you for your comment. We have removed the detailed description. 

Comment 16: Line 420 onward: The inclusion of sections on data transformations, missing data handling, and validation seems more aligned with quantitative meta-analysis than a scoping review. Please clarify their relevance in this context or consider omitting them.

Response 16: Thank you for your comment. The data transformation section has been removed from the protocol.

Comment 17: Line 506: There appears to be an inconsistency regarding quality assessment. Earlier, you mentioned quality considerations, but here it suggests there may be none. Please clarify your approach and consolidate all quality-related discussions into one section for clarity.

Response 17: Thank you for your observation regarding the quality assessment section. Upon careful consideration, we have decided to remove this section from our protocol. As per established methodological guidance, quality assessment is not a mandatory component of scoping reviews, which primarily aim to map existing literature and identify knowledge gaps rather than evaluate the quality of evidence. This approach aligns with frameworks such as Arksey and O'Malley's and the PRISMA-ScR guidelines, which consider critical appraisal optional in scoping studies. Therefore, we have omitted the quality assessment to maintain methodological consistency and focus on the objectives of our scoping review. We have removed the subheadings "Risk of Bias and quality consideration" and "Consideration of study quality " from the protocol. However, we have retained a brief statement on page 12, lines 516–4519, clarifying that a formal quality assessment will not be conducted in this review. 

Comment 18: Line 516: Stakeholder consultation is a key part of your approach. Consider referencing this earlier in the methods section.

Response 18: Thank you for your comment. We referenced the stakeholder consultation in the methods when we introduced the six stages that will guide the review.  This is on Page 4, line 123.

Comment 19: Additional Suggestions: Consider reducing the overall use of emphatic adverbs and adjectives throughout the manuscript to maintain a more scientific tone.

Response 19: Thank you for your comment. We have revived the manuscripts.  

Comment 20: If it does not restrict the scope, consider specifying “sub-Saharan Africa” in the title for clarity.

Response 20: Thank you for the comment. Our scope is Africa as it reflects in our tittle. 

Comment 21: Line 638: Begin the sentence with a capital letter.

Response 21: Thank you for the comment. WE have revised and the sentence now begin with a capital letter. Page 15, line 650.

Comment 22: Line 643: An isolated quotation mark appears here — please remove.

Response 22: Thank you quotation marks removed. Page 15, line 655.

Comment 23: Table A2: Please clarify if your search strategy will include all African countries listed individually. The current strategy may inadvertently exclude relevant studies from specific countries

Response 23: Thank you for your comment. To ensure comprehensive coverage, our search strategy will systematically include all African countries by listing them individually in the search terms. Recognizing the limitations of database query lengths, we will execute the search iteratively, incorporating a subset of countries in each run until all have been included. This approach aligns with established practices in systematic reviews focusing on African contexts. We will also document the specific search strings and iterations used to maintain transparency and reproducibility.​

Round 2

Reviewer 3 Report

Comments and Suggestions for Authors

Thanks to authors for considering the review comments .